# The Role of Apoptin in Chicken Anemia Virus Replication

**DOI:** 10.3390/pathogens9040294

**Published:** 2020-04-16

**Authors:** Cynthia Feng, Yingke Liang, Jose G. Teodoro

**Affiliations:** 1Department of Biochemistry, McGill University, Montreal, QC H3G 1Y6, Canada; 2Goodman Cancer Research Centre, Montreal, QC H3G 1A1, Canada

**Keywords:** chicken anemia virus, gyrovirus, apoptin, cell cycle, virus–host interactions

## Abstract

Apoptin is the Vp3 protein of chicken anemia virus (CAV), which infects the thymocytes and erythroblasts in young chickens, causing chicken infectious anemia and immunosuppression. Apoptin is highly studied for its ability to selectively induce apoptosis in human tumor cells and, thus, is a protein of interest in anti-tumor therapy. CAV apoptin is known to localize to different subcellular compartments in transformed and non-transformed cells, depending on the DNA damage response, and the phosphorylation of several identified threonine residues. In addition, apoptin interacts with molecular machinery such as the anaphase promoting complex/cyclosome (APC/C) to inhibit the cell cycle and induce arrest in G2/M phase. While these functions of apoptin contribute to the tumor-selective effect of the protein, they also provide an important fundamental framework to apoptin’s role in viral infection, pathogenesis, and propagation. Here, we reviewed how the regulation, localization, and functions of apoptin contribute to the viral life cycle and postulated its importance in efficient replication of CAV. A model of the molecular biology of infection is critical to informing our understanding of CAV and other related animal viruses that threaten the agricultural industry.

## 1. Chicken Anemia Virus

Chicken anemia virus (CAV) is the causative agent of chicken infectious anemia, a disease that poses a major economic problem in the global poultry industry [1]. CAV infection in young chickens results in severe immunosuppression and anemia caused by the viral destruction of cortical thymocytes and erythroblasts [2,3]. CAV causes high mortality in young chickens, especially in combination with other pathogens, due to its ability to induce immune dysfunction [2]. 

CAV is a non-enveloped single-stranded circular DNA virus of the genus *Gyrovirus* of the family *Anelloviridae* and is about 25 nm in diameter [4,5]. The CAV genome is circular and 2.3 kb in size, with a promoter region containing numerous 21-bp direct repeats. CAV produces a single polyadenylated polycistronic transcript with three overlapping open reading frames encoding the viral proteins, Vp1, Vp2, and Vp3 [6]. Vp1 is the only structural protein and constitutes the viral capsid [7]. Vp2 is a dual-specificity protein phosphatase that is likely to have a multifunctional role in viral replication and as a scaffolding protein for virion assembly [8,9]. Vp3 is a virulence factor that induces apoptosis in susceptible chicken lymphoblastoid T and myeloid cells [6].

Since the natural target of CAV in chickens are rapidly dividing cells, CAV has likely evolved mechanisms for replication in the division phase of the cell cycle. Whereas other DNA viruses such as adenovirus and papillomavirus often induce S phase in the host cell, likely to promote DNA replication including that of the viral genome, CAV induces cell cycle arrest in the G2/M phase [10,11,12]. This interaction with the cell cycle might suggest a rationale for why CAV preferentially infects transformed or otherwise rapidly-dividing cells, which are more likely to be undergoing DNA synthesis [13,14]. For example, CAV has been found to co-infect with other oncogenic DNA viruses that induce DNA replication, such as Marek’s Disease Virus (MDV) [15]. 

The ability of Vp3 to induce apoptosis became of particular interest when it was discovered that the effects were selective for tumor cells when introduced into non-chicken (i.e., non-natural host) cells, lending it the name “apoptin” [14]. Apoptin has subsequently been shown to induce apoptosis in many human cancer cell lines in a manner that is independent of p53 [14,16]. Apoptin’s tumor-cell killing ability has generated significant interest for its potential in anticancer therapy, and indeed apoptin has been investigated extensively in a wide range of human tumor cells, both in vitro and in vivo in mice, including melanoma, hepatoma, osteosarcoma, lung carcinoma, breast, and prostate cancers [17,18]. Additionally, the potential of apoptin as a gene therapy has been tested using an adenoviral vector and has been observed to reduce tumors in mice without significant side effects, showing its promise [19,20]. Characterization of apoptin as a cancer therapeutic continues to be widely researched and reviewed [21,22].

In transformed cells, apoptin inhibits the anaphase promoting complex/cyclosome (APC/C) through association with the APC1 subunit, causing cell cycle arrest in G2/M and inducing cell death through apoptosis [12]. Apoptin and the associated APC/C are targeted to promyelocytic leukemia (PML) bodies within the nucleus [23]. As the APC/C regulates entry into anaphase during mitosis, its inhibition stalls the cell cycle in G2/M, during which cells are known to globally downregulate translation [24]. In the case of many other viruses, inhibition of cap-dependent translation serves to cripple the host antiviral response and favor viral protein synthesis [25,26]. CAV is one of several viruses known to inhibit the APC/C [26], and viral inhibition of this complex proposes an interesting situation for the life cycle and replication of CAV and other viruses, which induce G2/M arrest.

Although apoptin has been investigated extensively for both its tumor-specific killing abilities and its pathogenesis in chicken thymocytes [14,27], the role of the protein in viral replication remains unresolved. Despite a significant body of literature on CAV and apoptin, many aspects of how the virus replicates in an infected cell remain unclear, particularly in terms of molecular biology. The purpose of this review is to examine the role of the apoptin protein in the context of the CAV life cycle and interpret how the functions of apoptin on the host cell might contribute to the virus’ ability to replicate.

## 2. Apoptin Structure and Regulation

### 2.1. Structure of Apoptin

The CAV Vp3 protein, also known as apoptin, is 121 amino acids in length and has a mass of 13.6 kDa [6]. While no 3D structure has been solved to date, the structure of apoptin has been studied extensively through the mutation of various residues in domains deemed to be important for cellular localization, protein–protein interactions, and apoptotic function (Figure 1).

Apoptin contains a bipartite C-terminal nuclear localization signal (NLS) from residues 82–88 and 111–121, which permits its translocation to the nucleus in a cell-type-specific manner; namely, it exhibits nuclear localization in transformed cells while remaining in the cytoplasm in non-transformed cells [28]. Mutation of either of these regions inhibits nuclear localization and consequently abolishes cell-type specific activity [23]. Interestingly, the particular NLS sequence of apoptin is important to the protein’s cell-type-specific function, as the region overlaps with the APC1 interaction domain [12,23]. When the apoptin NLS was replaced with that of another virus, the SV40 Large T NLS, cell-specific localization was preserved but apoptotic activity did not occur [23]. This demonstrated that nuclear localization of apoptin, while necessary, is insufficient for its activity.

Apoptin also possesses a leucine-rich Crm1-dependent nuclear export signal (NES) in the N-terminus, from amino acids 33–46, which permits nucleocytoplasmic shuttling [23,28]. Mutation in this N-terminal region removes the cell-type specificity of apoptin’s localization, wherein the mutated apoptin localizes to the nucleus in both transformed and non-transformed cells [23]. Though assumed to be a canonical Crm1-dependent NES, replacement of the apoptin NES with a prototypical, well-established NES of another protein, HIV-1 Rev NES, abolished the cell type-specific localization of apoptin, resulting in a diffuse cytoplasmic localization of apoptin in both transformed and non-transformed cells [23]. This finding determined that the specific sequence of the apoptin NES is critical to the cell-specific function of apoptin’s nuclear export.

The NES also has a partial sequence overlap with the apoptin multimerization domain found in the N-terminus (amino acids 1–48), which contributes to the observed phenomenon of apoptin aggregation in vitro and in vivo [23,29]. In cells where apoptin is localized to the nucleus, aggregation of apoptin via this domain obscures the NES, preventing nuclear export [6,28]. In this scenario, apoptin remains in the nucleus through its C-terminal NLS [28].

The apoptin NES region has additionally been shown to target apoptin and the associated APC/C to PML bodies within the nucleus [23]. Targeting to PML bodies is typically regulated by sumoylation of lysine residues, which has been confirmed for apoptin [30]. However, the precise lysine residues that are sumoylated in apoptin have not been identified, though residues in the leucine-rich sequence of the NES are required for targeting the PML bodies [30]. Apoptin was shown to bind directly to PML protein, although PML was not required for apoptotic activity [30]. The PML bodies are the target of many cellular and viral sumoylated or otherwise post-translationally modified proteins, and have established roles in tumor suppression, senescence, and most critically in apoptosis [23,31,32,33]. As many other DNA viruses have been found to interact with PML bodies, this association suggests that localization to nuclear PML bodies might play a role in promoting efficient viral replication [34]. Furthermore, apoptin’s targeting of the APC/C to PML bodies suggests modification of the APC/C’s mitotic functions and inhibition of APC/C activity, which could serve to facilitate apoptotic programming [23,35].

### 2.2. Localization and Regulation of Apoptin

The most studied function of the apoptin protein is its ability to induce apoptosis. This effect is dependent on apoptin’s translocation to the nucleus, which occurs only in transformed cells [28]. It has been established that cellular localization of apoptin is regulated by phosphorylation on various sites in the N- and C-terminal regions of the protein [36,37,38]. The first to be identified and subsequently most well-characterized is threonine-108 (T108), whose phosphorylation, though not essential, plays a role in the efficiency of apoptin activity and its cell type-specific localization [36,37]. The adjacent T107 was later found to also be phosphorylated, and in the event of T108 dephosphorylation, T107 serves as a compensatory phosphorylation site and results in a similar phenotype to T108-phosphorylated apoptin [39,40]. Additional phosphorylation sites of importance have been identified on apoptin at T56 and T61, in the N-terminal region [38]. Mutation of either of these threonines to alanine significantly impaired nuclear localization of apoptin and consequently attenuated viral production and cytopathic effect [38].

Several kinases have been proposed to phosphorylate apoptin through a variety of mechanisms. Upon induction of apoptosis by apoptin, protein kinase B/Akt translocates to the nucleus where it serves to promote apoptosis, contrary to its normal pro-survival function, via cyclin-dependent kinase 2 (Cdk2) as an effector [41]. Using in vivo techniques of Cdk2 inhibition with both the compound roscovitine and Cdk2-directed siRNA, Maddika et al. (2009) confirmed that Cdk2 phosphorylates apoptin, though whether the target residue was T108 could not be identified [41]. Another cell cycle regulatory kinase, Cdk1, has also been implicated in apoptin function, as Cdk1 knockdown resulted in an impaired ability of apoptin to localize and induce apoptosis in transformed cells [42]. Additionally, protein kinase Cβ has also been shown to mediate phosphorylation state and nuclear localization of apoptin in tumor cells [43].

Though initial hypotheses proposed that T108 phosphorylation on apoptin is essential for its activity, this was later disproved by Guelen et al. (2004) and subsequently corroborated by Lee et al. (2007) [36,44]. Kucharski et al. (2011) further confirmed that T108 phosphorylation is not required for apoptin’s cell type-dependent localization [45]. Instead, they found that nuclear localization of apoptin occurs only during the DNA damage response (DDR) [45]. This was confirmed by two models of DDR activation—a cellular model, using Bub1 mitotic checkpoint kinase knockdown, and a chemical model, using bleomycin [45]. In both cases, apoptin localization to the nucleus correlated with the appearance of γH2AX, a major marker for DNA damage [45]. Further work from Kucharski et al. (2016) confirmed that DDR regulates apoptin nuclear localization and consequently its apoptotic effect, specifically through checkpoint kinases Chk1 and Chk2 [38]. Inhibition of Chk1/2 in transformed cells resulted in a cytoplasmic accumulation of apoptin, impairing its activity [38]. Chk1/2 were shown to regulate apoptin localization through phosphorylation of threonines T56 and T61, but not T108 [38].

This finding in the context of the virus offers insight into how a full activation of apoptin activity might occur. In addition to the regulation occurring via phosphorylation of apoptin in transformed cells, apoptin necessarily translocates to the nucleus upon activation of the DDR, which can also be triggered by viral replication of its single-stranded DNA genome in non-transformed infected host cells [28,38,46]. DDR signaling events would lead to an abundance of activated Chk1/2 and phosphorylated apoptin, similarly to that in transformed cells, allowing for apoptin’s translocation to the nucleus, productive viral replication, and death of the host cell [38].

As apoptin possesses nucleocytoplasmic shuttling ability, its dephosphorylation must also be a regulated process. Based on observations of Vp2 and T108-phosphorylated apoptin co-localizing in the cell and that of Vp2 association to apoptin in pulldown assays, the putative phosphatase of the T108 on apoptin is the virus-encoded Vp2 dual-specificity phosphatase [47]. Vp2 dephosphorylation of T108 appears to downregulate the apoptin-induced cytopathic effect but does not completely abolish it, and this function seems to play a role in modulating the CAV infection process [47]. It remains to be determined whether other phosphorylation sites on apoptin are similarly regulated by Vp2 or whether other phosphatases might be involved, and to what extent the various phosphorylation sites, kinases, and phosphatases exhibit control over apoptin’s function and activity.

## 3. Role of Apoptin in CAV Infection

### 3.1. Apoptin and the APC/C

The anaphase promoting complex/cyclosome (APC/C) is a 1.2 MDa protein complex that consists of more than 15 different subunits [48]. The APC/C is a RING-family E3 ubiquitin ligase that along with the SCF (Skp, Cullin, F-box) complex (another ubiquitin ligase), is one of two central regulators of the cell cycle [48,49,50]. In particular, the APC/C regulates the cell cycle from mitotic entry until the end of the G1 phase [50]. Activation and specific activity of the APC/C relies on association with specific coactivators, depending on the cell cycle stage—Cdc20 during mitosis and Cdh1 during G1 [50]. Coactivators are associated with the APC/C through three conserved sequence motifs—the C box and the KILR tetrapeptide motif, both found in the N-terminal domain, as well as the IR dipeptide tail, found in the C-terminus [51]. Despite its large size and complicated arrangement, only 2 subunits, APC2 and APC11, are catalytic modules [48]. The remaining subunits function in either the recognition of and interaction with substrates and coactivators (APC1, APC3, and APC8), or as scaffolding [48,52].

The primary substrate of the APC/C’s E3 ligase activity is securin [50]. Securin is a small protein and the inhibitor of separase, a protease that separates sister chromatids during mitosis. During metaphase, the APC/C ubiquitinates securin, which targets it for degradation, freeing separase to promote the separation of sister chromatids, and consequently moving the dividing cell from metaphase to anaphase [50]. Being a master cell cycle regulator, the APC/C also targets cyclin A, cyclin B, and downstream cell cycle regulators, depending on the specific stage of the cell cycle [50].

Apoptin interacts with the APC/C by binding the APC1 scaffolding subunit of the complex [12]. Unfortunately, a lack of structural data on apoptin makes it difficult to suggest specific molecular interactions. Various residue mutations in the C-terminal domain of apoptin, including those of K116, R117, and R118 to alanine, have been shown to abolish apoptin’s association with the APC1, identifying the C-terminus as essential for APC1 binding [12]. In addition, Heilman et al. (2006) have identified that the sequence that serves as the NLS in the C-terminus also seems to be essential for interaction with the APC/C [23]. Apoptin interaction with APC1 leads to inhibition and disassembly of the APC/C complex, resulting in host cell G2/M arrest [12]. It has been postulated that this interaction of apoptin with the APC/C is the purpose of apoptin’s nucleocytoplasmic shuttling activity, in order to allow recruitment of cytoplasmically localized APC/C to the nucleus, and to subsequently target it to the PML nuclear bodies [23,53]. This could serve to facilitate the viral life cycle, as the sequestration and inhibition of the APC/C along with induced mitotic arrest might trigger apoptotic signaling or play other functional roles for the CAV life cycle.

### 3.2. Apoptin and G2/M Cell Cycle Arrest

The G2 phase of the cell cycle is maintained by the Wee1–Cdc25 complex while regulation of mitotic entry is governed by the cyclin B/Cdk1 complex (Figure 2) [54,55]. As the G2 phase progresses, cyclin B levels begin to accumulate, reaching peak levels by early mitosis [55]. Mitotic progression and exit are in turn regulated by the degradation of cyclin B, which is mediated by the E3 ubiquitin ligase activity of the APC/C [50,55]. As the G2 phase is the last phase of the cell cycle before mitosis, it contains the final checkpoint for detection of DNA damage and genomic integrity before the cell undergoes division [55]. Activation of this checkpoint causes the cell to arrest in G2/M and triggers a corresponding cellular response until the problem has been resolved [56]. If the cell stays in the G2/M stage for a prolonged period, apoptosis tends to occur, a feature that has been exploited by several tumor-targeting chemical compounds [57,58,59,60,61].

In G2/M phase, cap-dependent translation is highly downregulated to approximately 25% the rate of translation of cells in interphase [62]. This begs the question of why a virus would seemingly induce and favor G2/M arrest. In addition to CAV, several other viruses are known to induce G2/M arrest in infected cells, including human immunodeficiency virus 1 (HIV-1) and infectious bronchitis virus (IBV). HIV-1 encodes a conserved protein, Vpr, that induces a cell cycle block at G2/M, during which its transactivator, Tat, appears to hijack the cellular machinery to favor viral transcription and promote efficient viral replication [63,64]. IBV, an avian virus of the coronavirus family, similarly induces G2/M arrest in host cells [65]. IBV protein synthesis was found to be upregulated during this phase, compared to that in other phases, contradictory to the regular observation that cap-dependent translation is downregulated in the G2/M phase, which suggests that IBV translation might be enhanced by a cellular factor optimally expressed in G2/M or might occur in a cap-independent manner [62,65]. It has also been postulated that cell cycle arrest before mitosis might serve to prevent disruption of the Golgi apparatus and endoplasmic reticulum, to favor assembly of viruses that use these structures, which include the coronaviruses [65,66,67,68].

In some cases, viral protein synthesis might be under the control of an internal ribosomal entry site (IRES), for which an advantage might exist during the cellular global downregulation of cap-dependent translation in the G2/M phase [65,69,70,71,72]. During this phase, IRES-dependent translation of certain cellular and viral mRNAs has been found to be optimal, while translation of other IRES-dependent mRNAs might actually be downregulated, such as in the case of coxsackievirus [73]. Other transcripts have even been observed to undergo cap-independent translation only in specific cell cycle stages, such as in G2/M, while their translation in the G1/S phase remains cap-dependent [74]. This model could be applied to viral regulation, where certain viral proteins might be translated over others, depending on the phase of the host’s cell cycle. This has been observed in certain coronaviruses including IBV and mouse hepatitis virus, where some mRNAs are translated in a cap-dependent manner while others are cap-independent [65].

Based on the context of other viruses that induce G2/M arrest, there is a possibility that CAV uses G2/M arrest as a means of either upregulating virus translation and replication through hijacking of cellular factors present in the G2/M phase or as a means of regulating protein expression. There has been no conclusive evidence for or against whether CAV could undergo cap-independent translation or whether the CAV mRNA possesses an IRES and hence this could be an area of interest.

The G2/M phase might serve further as a means of facilitating viral replication by downregulating the interferon response. Recently, Bressy et al. (2019) investigated how the antiviral response differs depending on the stage of the cell cycle using mutated vesicular stomatitis virus (VSV), a widely used model for studying virus–host interactions, in cells arrested at various stages in the cell cycle using chemical compounds [75]. Interestingly, they found that during the G2/M phase, the antiviral response is inhibited [75]. In particular, the antiviral interferons and the associated interferon-stimulated genes were inhibited, which greatly enhanced the efficiency of viral replication and secondary infection [75]. This finding could provide an immunological rationale for why viruses induce G2/M arrest, in addition to the differences in host cell activity at the cellular level.

### 3.3. Apoptin in the Viral Life Cycle

Ultimately, apoptin is a virulence factor in CAV and is one of only three proteins encoded by the virus [6]. This simplicity of the virus suggests that CAV must be very efficient and that each protein is likely to be multifunctional, in order to aid and facilitate viral replication. In order to understand apoptin’s role in viral functions and propagation, it is also necessary to examine the viral life cycle as a whole. 

Electron microscopy of CAV in the cell shows that it largely accumulates in the nuclei of infected cells [76]. Vp1, the CAV capsid protein, contains both NLS and NES sequences, providing evidence that CAV is assembled in the nucleus [77,78]. Whether apoptin interacts with Vp1 is yet to be determined, though apoptin has been observed to interact with Vp2, and Vp2 is similarly known to interact with Vp1 [47,79]. Sun et al. (2018) recently detected that distinct sites on Vp2 are required for interaction with Vp1 and apoptin, suggesting that Vp2 might link their biological functions and might also have a role as a scaffolding protein [80]. As presence of CAV particles have been observed in apoptotic bodies of infected dying cells that were then absorbed by neighboring epithelial cells, it could be deduced that the cytopathic effect of apoptin is important to the pathogenesis of the virus [27]. This process provides a suitable model for non-enveloped viruses, such as CAV, to induce apoptosis as a means of viral egress.

Taken together, we can posit a model for CAV infection and viral propagation (Figure 3). Upon entry of the virus into a cell, its ssDNA genome enters the nucleus and is transcribed. Then, as mRNA translation occurs, apoptin is made, and in transformed cells or cells with activated DDR, apoptin binds the APC/C, localizes to the nucleus and induces G2/M arrest, which promotes efficient viral replication. Similarly, the Vp1 and Vp2 proteins also accumulate, and facilitate viral particle assembly in the nucleus. Apoptosis of the host cell then promotes viral egress, and CAV sequestered into apoptotic bodies may be phagocytosed by neighboring cells, providing a means of entry and continued infection.

## 4. Apoptin-Like Proteins in Other Single-stranded DNA Viruses

### 4.1. Comparing Apoptin to Similar Viral Proteins

In studying the role of a viral protein on the virus’ capability to replicate and propagate as a whole, it is often valuable to examine mechanistically similar viral proteins or related viral families, whose study could provide insight into the purpose of the mechanisms.

Apoptin does not possess any significant sequence or structure homology to any known host or viral proteins. Determination of the structure of apoptin has not been achieved, largely due to the tendency of apoptin to form multimeric aggregates both in vitro and in vivo, as well as the lack of secondary structure within aggregated apoptin [29]. However, using homology modeling, Panigrahi et al. (2012) succeeded in constructing a 3D-model of monomeric apoptin, through the combination of peptide sequences from numerous proteins with regions similar to those in apoptin [81].

Since such a model does not necessarily rely on viral proteins or proteins with any functional similarities to apoptin, we instead examined similarities between apoptin and other viral proteins found in related single-stranded DNA viruses. Namely, we discuss proteins found in other members of the anellovirus and circovirus families that have been observed or hypothesized to have similar functions to apoptin. Comparison with other single-stranded DNA viruses that encode apoptin-like proteins suggests that the role of CAV apoptin in viral replication and pathogenesis is likely critical, and that the interactions with host-cell factors might be conserved within related viral families.

### 4.2. Anelloviruses

CAV was recently reclassified to the *Anelloviridae*, due to closer similarities than to its previous viral family of *Circoviridae* [5]. The type virus of the anelloviruses, torque teno virus (TTV), is widespread in the human population, replicating in liver cells, though its pathogenic effects remain unclear [82]. TTV and CAV share comparable genomic organizations, with both harboring negative-sense single-stranded DNA genomes containing overlapping open reading frames (ORFs) [83,84]. CAV and TTV share several conserved motifs between their Vp1 and ORF1, respectively, as well as another highly conserved motif between the CAV Vp2 and TTV ORF2 [84]. The functions of each of these proteins are also similar—the TTV ORF1 and CAV Vp1 are structural proteins in the capsid, the TTV ORF2 and CAV Vp2 both possess phosphatase activity, and most notably, the TTV ORF3, like apoptin, has been shown to have apoptotic activity in certain cancer cell lines [82].

In a study by Prasetyo et al. (2009) where cells were transfected with an apoptin-knockout clone of CAV, impaired DNA replication was rescued not only by the supplementation of wild-type apoptin, but also by the TTV ORF3 [85]. These results provided further evidence for a close-relatedness of CAV and TTV and their Vp3/ORF3 proteins, as they not only share similarities in terms of apoptotic oncolytic activity, but likely also in viral replication [85]. This work further highlights that the apoptotic activity of CAV and other viruses is useful and important to the viral life cycle, and is not primarily a process for disease pathogenesis.

A more recently discovered virus of the *Gyrovirus* genus, human gyrovirus (HGyV), is a more closely related virus to CAV, with similar genomic structure encoding three proteins, Vp1, Vp2, and Vp3 [86]. Similar to apoptin, studies on the HGyV Vp3 protein have shown that the protein also induces G2/M arrest and apoptosis in transformed cells [87,88]. HGyV Vp3 displayed similar subcellular localization patterns to apoptin, wherein it translocated to the nucleus in cancer cells but remained predominantly in the cytosol in non-transformed cells [87]. This early data suggests that HGyV Vp3 might retain the mechanism of tumor-cell-specific activity seen with apoptin and provides support for an apoptin-like protein playing a conserved critical role in the natural viral life cycle of gyroviruses, as well as other anelloviruses.

### 4.3. Circoviruses

CAV does not exhibit a close phylogenetic relationship to the circoviruses and studies have demonstrated that gyroviruses such as CAV are not structurally related to circoviruses [77,84]. However, despite CAV having been reclassified away from the *Circoviridae*, there are some notable similarities in terms of viral protein functions and the viral life cycle, which might emphasize the importance of these viral mechanisms. The *Circovirus* genus of the *Circoviridae* family contains important avian and porcine pathogens, including beak and feather disease virus (BFDV), and porcine circovirus 2 (PCV-2) [89]. In contrast to CAV and the gyroviruses, circoviruses have a circular genome with an ambisense organization, containing at least two identifiable ORFs, a conserved replication-associated protein (Rep), and a capsid protein (Cp) [89]. In addition to the Rep- and Cp-encoding ORFs, circoviruses may express other proteins—for example, PCV-1 and PCV-2 are known to encode a third and fourth protein, ORF3 and ORF4, with apoptotic activity and potential anti-apoptotic functions, respectively [90,91,92,93].

In the case of PCV-2, a circovirus that causes postweaning multisystemic wasting syndrome in pigs, the ORF3 has been shown to induce apoptosis through a caspase-8 dependent pathway [92]. This protein appears to function by a different mechanism than CAV apoptin, which has been shown to induce apoptosis independent of caspase-8, and instead has been suggested to mediate apoptosis through the mitochondrial cell-death pathway [94]. Though PCV-2 ORF3 has been observed to induce apoptosis in melanoma cells, both in vitro and in a mouse model, demonstrating potential as a cancer therapeutic, the effect on tumor reduction is less dramatic than that of apoptin [19,95]. Additionally, PCV-2 ORF3 does not cause G2/M arrest in infected cells and similar apoptotic effects were observed in both non-transformed and transformed cells, suggesting that PCV-2 ORF3 does not share the tumor-cell-specific killing of apoptin [92,95]. Although differences in tumor selectivity of PCV-2 ORF3 and CAV apoptin might exist, ORF3 has interestingly been demonstrated to play a role in systemic dissemination of the PCV-2 infection, by expediting the viral spread [96]. These data further support that apoptin-like proteins appear to aid in the viral life cycle of CAV-related viruses and lend themselves to efficient viral replication.

Another porcine circovirus, PCV-1, also encodes a viral protein with apparent tumor-selective apoptotic properties [91]. Though the apoptotic activity of apoptin seems directly linked to its nuclear localization and occurs only in transformed cells, PCV-1 ORF3 appears to localize the cytoplasm in both normal and transformed cells, but displays apoptosis-inducing activity only in transformed cells [91]. Unlike PCV-2, PCV-1 is non-pathogenic and ubiquitous in pigs; however, PCV-1 ORF3 exhibited more induction of apoptosis in more types of cells than PCV-2 ORF3 [97]. Assuming homology in viral protein functions due to their similarity in structure, these findings suggest that the ORF3 of porcine circoviruses is not the sole determinant of pathogenicity in pigs and might instead point to a functional role during viral replication.

## 5. Conclusions

Though apoptin’s functional effect on cells remain of particular interest as a tool in cancer therapy, a molecular model of apoptin’s interactions within the cell has yet to be understood in the context of CAV’s natural course of infection. In this review, we interpreted data obtained from a variety of purposes for studying CAV to construct a model of the viral life cycle in which apoptin plays an integral role in efficient viral replication, propagation, and egress. Coordination with other viral proteins to induce cellular responses such as the DNA damage response appears to be particularly important for apoptin to exert its pro-viral functions. Questions remain surrounding the exact mechanism of how apoptin’s interaction with the APC/C triggers apoptosis, as well as the specific regulation of various phosphorylation sites on apoptin and how they coordinate to contribute to apoptin’s activation in either transformed or non-transformed susceptible host cells. Further analysis would also be required to determine whether models describing how other viruses might induce G2/M arrest apply to CAV, in order to comprehensively understand how CAV manipulates the cell cycle to replicate so efficiently and continues to pose a persistent threat to poultry worldwide. 

## Figures and Tables

**Figure 1 pathogens-09-00294-f001:**
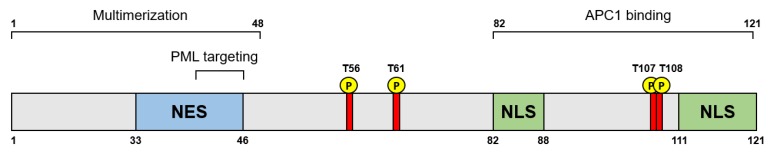
Schematic of the chicken anemia virus (CAV) Vp3 protein, apoptin. A leucine-rich nuclear export signal (NES) is identified (blue) in the N-terminus from amino acids 33–46, and a bipartite nuclear localization signal (NLS) is pictured (green) in the C-terminus from amino acids 82–88 and 111–121. A region from amino acids 1–48 containing a multimerization domain and sequences in the NES that are important for targeting to promyelocytic leukemia (PML) bodies are indicated with brackets. The APC1 binding domain from amino acids 82–121 in the C-terminus is also shown. Threonine residues at positions 56, 61, 107, and 108 are labelled (red) as important phosphorylation sites for regulation.

**Figure 2 pathogens-09-00294-f002:**
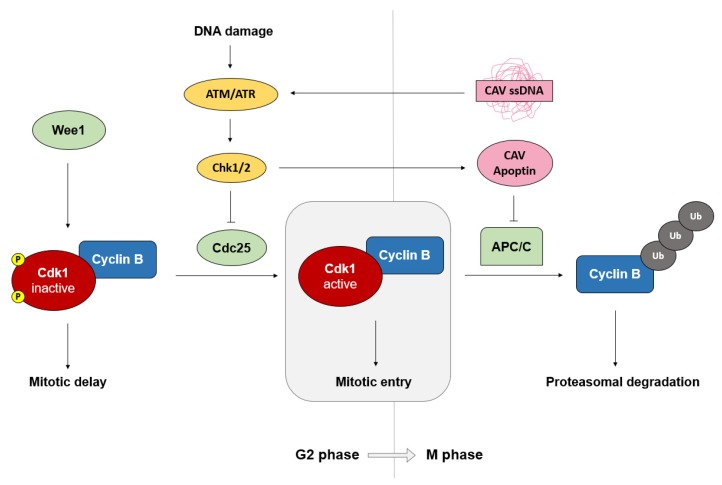
Schematic of the G2/M phase checkpoint mediated by cyclin B/Cdk1. Wee1 kinase phosphorylates Cdk1 to maintain a cell in G2 phase and delay mitosis, while Cdc25 phosphatase removes inhibitory phosphates from Cdk1, allowing its active form and promoting entry into mitosis. Activation of the DNA damage response via ATM (ataxia telangiectasia mutated)/ATR (ATM- and RAD3-related) triggers checkpoint kinases Chk1/2 to inhibit Cdc25, preventing activation of the cyclin B/Cdk1 complex and delaying M phase entry. During M phase, the APC/C ubiquitin (Ub) ligase polyubiquitinates cyclin B for proteasomal degradation, regulating mitotic progression and exit. Upon CAV infection, single-stranded genomic viral DNA is sensed by ATM/ATR, activating Chk1/2, which phosphorylate CAV apoptin. Phosphorylated apoptin localizes to the nucleus where it inhibits the APC/C and prevents continuation of the cell cycle.

**Figure 3 pathogens-09-00294-f003:**
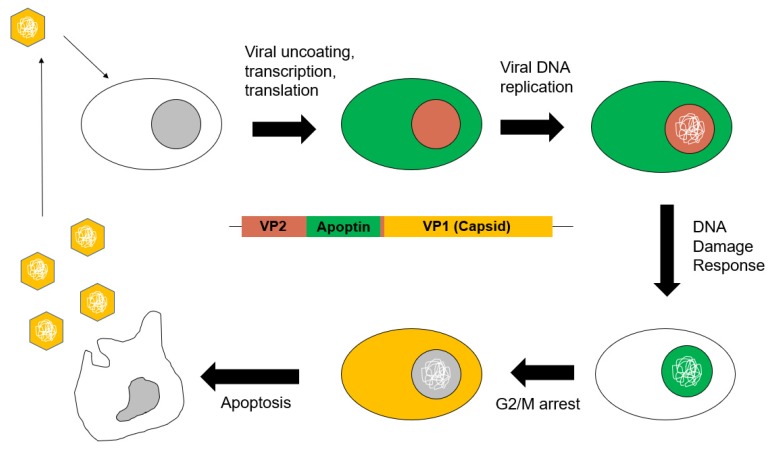
Schematic diagram showing the proposed model of the CAV life cycle. Entry of the virus into the cell is followed by viral uncoating, transcription, and translation. The putative replicase, Vp2 (pink), would localize to the nucleus to initiate viral DNA replication while apoptin (green) is initially localized in the cytoplasm. As viral DNA accumulates and triggers the DNA damage response, apoptin is phosphorylated and localizes to the nucleus, where it aggregates and sequesters the APC/C. G2/M phase arrest is induced while viral capsid protein (yellow) and viral genome continue to be synthesized. Virions (yellow hexagons) are packaged and apoptosis induced by the apoptin-mediated G2/M arrest facilitates viral egress and spread.

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
