# Peer review of "The Role of Apoptin in Chicken Anemia Virus Replication"

_pathogens, 2020, doi:10.3390/pathogens9040294_

Round 1

Reviewer 1 Report

Manuscript Title: The role of apoptin in chicken anemia virus replication

General comments:

In the current review the authors have focused on the role of the protein Apoptin in chicken anemia virus (CAV) replication. This is a very well written review article where the authors have meticulously focused on its actual functional role within the virus rather than it anti-cancer properties. I liked the section where the authors have focused on similarities with other family of viruses which I found to be very informative. I therefore would like to see this article published after the following minor corrections.

Comments:

  • In fugure1.the authors if possible should present the model of this protein that is based on homology modeling.
  • A sequence alignment of Apoptin/ or a table with sequence similarity and identity with similar proteins from other viruses should be presented as a separate figure/table.

Author Response

We thank the reviewer for the comments and suggestions to our manuscript. We agree that a structural representation of the protein apoptin based on homology modeling would be very useful; however, we feel that homology modeling is unsuitable in this instance due to a lack of information on the structure of apoptin or any similar proteins. In sequence alignment searches (BLASTp), we are unable to identify any other proteins with high similarity, with only small similarities with closely-related Vp3 homologs of avian and human gyrovirus isolates. In our attempt to use homology modeling and secondary structure prediction for apoptin, our result was simply a largely disordered protein was no clear identifiable domains of known structure. Therefore, we were unable to make the reviewer’s suggested changes to Figure 1. However, a further search in the literature following the reviewer’s suggestions helped us find one group’s published model of apoptin done with homology modeling, which we have now added and referenced in the text (Lines 309-311).

The similarities between apoptin and other proteins we cover in Section 4 (found in other single-stranded DNA viruses) are based on functional similarity rather than sequence identity or similarity; thus, we did not include a figure or table to show these alignments. An added subsection (new Section 4.1, beginning on Line 302) introduces an overview and clarifies that the similarities we examine are based on protein functional role in a virus rather than structural homology. 

Reviewer 2 Report

The manuscript is well written and approaches deeply, as well as we can read in the text submitted to the Pathogens journal. There is only a point to be corrected: in the abstract - line 12 - where is read "However... remains unknown". In my opinion, this sentence should be removed, because it seems out of context and it is not adding a relevant information to the review. Certainly, this review will contribute for increasing knowledge of CAV infection and role of apoptin in pathogenesis of infection.

Author Response

We thank the reviewer for their comments on our manuscript. The sentence in question (beginning on Line 12) was removed following the reviewer’s suggestion.

Reviewer 3 Report

This is a very well written review of the role of apoptin in Chicken Anemia Virus (CAV) replication. CAV causes chicken infectious anemia, a major food security problem. Unusually, CAV arrests cells in G2/M of the cell cycle. Apoptin has shown promise as an anticancer agent. The structure and function of apoptin in the CAV infectious life cycle is particularly well described. I have made a few suggestions for the authors to improve the manuscript.

  1. Figure 1 could be more informative. It is quite rudimentary at present. Adding in the other domains mentioned in the text e.g. the multimerization domain, the PML targeting domain etc., would be helpful, as would annotation with amino acid numbering. Addition to the diagram of the other threonines through which Chk1/2 regulates apoptin would also be helpful, or if this is too complex, listing these in a Table.
  2. A figure to illustrate the interactions discussed in the paragraph starting on line 199 would be helpful.
  3. The section on apoptin-like proteins in other ssDNA viruses is interesting but seems a little separate from the mechanistic discussions that precede it, perhaps due to lack of information for some of these viruses. The authors should consider including a section on apoptin as a cancer therapeutic.

Author Response

We thank the reviewer for their comments and suggestions to our manuscript. We have attempted to make slight improvements to Figure 1 based on the reviewer’s comments. Additional amino acid numbering has been added to show the positions of the signaling sequences and domains. The multimerization and PML targeting regions have also been indicated using brackets – however the exact sequences important to these functions are less clearly identified in the literature than the nuclear export and nuclear localization signals. The multimerization domain is known to be within the first 48 N-terminus residues (Heilman et al., PMID: 16840333) and PML targeting requires sequences in the leucine-rich NES (Janssen et al., PMID: 16924230). The APC1 interaction domain, in the C-terminal residues 82-121 (Teodoro et al., PMID: 15314021) was similarly added to the figure. In addition to these changes to the figure, wording was adjusted throughout Section 2 to clarify and reflect these points: addition of detail in Line 106-107; removal of misleading sentence in Line 112-113 and replacement with Lines 113-116; specification in Line 159-160. Regarding the last of these changes, while other threonine residues were studied for Chk1/2 phosphorylation, only T56 and T61 were found to be relevant in the regulation of apoptin localization (Kucharski et al., PMID: 27512067). Therefore no additional threonine phosphorylation sites were added to Figure 1 and the language was clarified in the text.

The reviewer’s suggestion of including a figure to illustrate the paragraph now starting on Line 210 was received. A new schematic showing background information of the G2/M checkpoint and how the CAV genome and its protein apoptin interact with it are now shown in a new figure, labelled Figure 2 (Lines 220-229). (The subsequent figure, previously labelled Figure 2, was changed to Figure 3.)

We acknowledge the reviewer’s comments that the section on apoptin-like proteins in other ssDNA viruses may seem separate from the other sections of this review. We have added a new subsection, now Section 4.1 (Lines 302-318), to summarize the importance of covering these similar proteins as well as provide an overview to how we focused our attention to particular proteins and viruses. The numbering of the subsequent subsections (now 4.2 and 4.3) were changed accordingly. Sentences in Lines 320-322 and in Lines 385-387 were removed as they were moved to subsection 4.1.

Regarding the reviewer’s suggestion to add a section about apoptin as a cancer therapeutic, our intent of the review is to focus on the role of apoptin within the virus rather than its potential as an anticancer agent. A number of reviews on apoptin’s potential in cancer therapy have already been published, such as by Los et al. (PMID: 19374922) and by Tavassoli et al. (PMID: 16133863). Nonetheless we made an effort to address the reviewer’s comment by adding more detail in Lines 50-57 as well as providing new references to comprehensive reviews as well as more recent work focused on apoptin as an anticancer agent.